# Unravelling the Role of P300 and TMPRSS2 in Prostate Cancer: A Literature Review

**DOI:** 10.3390/ijms241411299

**Published:** 2023-07-11

**Authors:** Charitomeni Gioukaki, Alexandros Georgiou, Lydia Evangelia Gkaralea, Christos Kroupis, Andreas C. Lazaris, Christos Alamanis, Georgia Eleni Thomopoulou

**Affiliations:** 1First Department of Pathology, School of Medicine, National and Kapodistrian University of Athens, 11527 Athens, Greece; charitomenig@med.uoa.gr (C.G.);; 2Department of Medical Oncology, Agii Anargiri Cancer Hospital, 14564 Athens, Greece; lydiagkaralea@gmail.com; 3Department of Clinical Biochemistry, Attikon University Hospital, National and Kapodistrian University of Athens, 12461 Athens, Greece; 41st Urology Department, Laiko Hospital, National and Kapodistrian University of Athens, 11527 Athens, Greece; 5Cytopathology Department, Attikon University Hospital, National and Kapodistrian University of Athens, 12461 Athens, Greece

**Keywords:** p300, TMPRSS2, androgen receptor, prostate cancer, molecular pathology, precision medicine, Gleason score

## Abstract

Prostate cancer is one of the most common malignant diseases in men, and it contributes significantly to the increased mortality rate in men worldwide. This study aimed to review the roles of p300 and TMPRSS2 (transmembrane protease, serine 2) in the AR (androgen receptor) pathway as they are closely related to the development and progression of prostate cancer. This paper represents a library-based study conducted by selecting the most suitable, up-to-date scientific published articles from online journals. We focused on articles that use similar techniques, particularly those that use prostate cancer cell lines and immunohistochemical staining to study the molecular impact of p300 and TMPRSS2 in prostate cancer specimens. The *TMPRSS2:ERG* fusion is considered relevant to prostate cancer, but its association with the development and progression as well as its clinical significance have not been fully elucidated. On the other hand, high p300 levels in prostate cancer biopsies predict larger tumor volumes, extraprostatic extension of disease, and seminal vesicle involvement at prostatectomy, and may be associated with prostate cancer progression after surgery. The inhibition of p300 has been shown to reduce the proliferation of prostate cancer cells with *TMPRSS2:ETS* (E26 transformation-specific) fusions, and combining p300 inhibitors with other targeted therapies may increase their efficacy. Overall, the interplay between the p300 and TMPRSS2 pathways is an active area of research.

## 1. Introduction

After lung cancer, prostate cancer is the second leading cause of cancer-related death in men, according to the latest data provided by Cancer Statistics 2022 [1]. There is a geographical variation in prostate cancer incidence. In developed countries, the disease is more prevalent than in developing countries. Various factors may contribute to these differences in incidence rates, including lifestyle, diet, genetics, and access to healthcare [2]. Our limited knowledge about the physical history of prostate cancer, as well as the unmapped molecular events occurring before and after hormone treatment, limits the therapeutic options. In the past few years, advances in molecular biology and the widespread use of sequencing techniques in medical research have transformed our understanding of this disease, paving the way to precision medicine [3,4].

There is evidence that androgen receptor (AR) co-activators are implicated in both androgen-dependent and androgen-independent prostate cancer [5,6]. The pathogenetic role of the AR pathway is evident in the majority of prostate cancer cases, both androgen-dependent and castration-resistant, as indicated by the continuously increasing levels of the prostate-specific antigen (PSA). Molecular studies of AR co-activators have been widely conducted to understand prostate cancer progression. There are some co-activators that play a tumor suppressor role, while others play a positive role in regulating cancer progression [7,8].

In this study, we sought to investigate the implication of p300 and TMPRSS2 proteins in the molecular pathology of prostate cancer. We performed a thorough library-based research of the currently published bibliographic evidence that aligns with our research objectives. For a detailed overview of our search strategy, please refer to Appendix A.These two molecules are being extensively studied for their role in carcinogenesis, disease progression, and metastasis.

## 2. Biological and Biochemical Features of TMPRSS2

### 2.1. Biological and Biochemical Features of TMPRSS2

The transmembrane protease serine 2 (TMPRSS2) is a member of the type II transmembrane serine protease (TTSP) family. The *TMPRSS2* gene is located on chromosome 21q22.3, and it is expressed in different anatomic locations in a time-preset manner, including in the fetal nervous system and the adult respiratory epithelium [9,10]. High expression is also marked in the prostate epithelial cells. The *TMPRRS2* gene harbors androgen-responsive elements in the 5′-UTR regions, placing its expression under hormonal control through the stimulation of the AR [9,11].

The advent of the COVID-19 pandemic brought TMPRSS2 into the spotlight of the scientific community due to its significant function in facilitating the entry of the Severe Acute Respiratory Syndrome Coronavirus 2 (SARS-CoV-2) and its variants into the host cells [12,13,14]. These scientific investigations have not only illuminated the comprehension of the pathogenic mechanisms of the virus that allow cell entrance, but they have also unraveled the molecular structure and certain biological functions of TMPRSS2. As a member of the TTSP family, TMPRSS2 is produced as a zymogen and undergoes post-transcriptional modifications to obtain its active structure. TMPRSS2 consists of three distinct regions, including an intracellular portion, a transmembrane domain, and an extracellular domain. The latter harbors the proteolytic function of the molecule, and it is in part highly conserved among TTSPs [15].

Two different isoforms of the TMPRSS2 protein, occurring through alternative splicing of mRNA, have been identified. Both isoforms share similar transmembrane and extracellular domains. Isoform 1 exhibits a longer N-terminal intracellular domain and contains 32 more amino acids than isoform 2. They are produced as zymogens, and they are autocatalytically activated [9,16]. The extension in isoform 1 is thought not to affect its efficiency in autocatalytic activation; however, it might influence cleavage specificity [16].

### 2.2. The Biological Role of TMPRSS2 in the Normal Prostate Gland

Although the abundant expression of TMPRSS2 is evident in normal prostatic tissue, its exact biological role is not clear. Different endogenous substrates of TMRPSS2 have been identified, including the epithelial sodium channel (ENaC) [17] the protease-activated receptor-2 (PAR-2) [18], as well as other serine protease-like zymogens, such as kallikrein-2 (KLK2) [19], suggesting its potential contribution to prostate homeostasis and its plausible involvement in male fertility. On the other hand, Kim et al. observed no phenotypic alterations resulting from the loss of TMPRSS2 expression in their study in murine animal models. By using homologous recombination, they disturbed the serine protease domain, generating Tmprss2−/− knockout mice. Compared to their wild-type counterparts, the Tmprss2−/− mice showed no phenotypic difference in terms of survival, normal prostatic development, and growth or fertility. Based on this finding, they suggested the presence of functional redundancy with other members of the TTSP family or the fact that the TMPRSS2 specialized role is evident under stressful situations or diseases [20].

### 2.3. The Role of TMPRSS2 in Prostate Carcinogenesis

TMPRSS2 is dysregulated in several types of malignant neoplasms, including breast, lung, gastric, ovarian, and renal carcinomas. There are diverse expression patterns within distinct tumor subtypes, as well as in comparison to the adjacent normal tissue parenchymal. In addition, there is a positive correlation between increased TMPRSS2 expression and the levels of immune infiltrates in breast cancer and lung adenocarcinoma [21]. Conversely, in a different study, *TMPRSS2:ERG* fusion detected by RNA sequencing in prostate cancer samples was associated with a lower lymphocytic infiltration, a finding that could potentially interfere with the biology of the tumor immune response [22]. The extent of immune cell infiltration in tumors is known to serve as a prognostic factor for survival [23,24]. As such, TMPRSS2 may serve as an important biomarker for the prognosis of patients with cancer.

The enzymatic activity of TMPRSS2 has also been implicated in cancer invasion. Ko et al. performed in vitro studies in three different prostate cancer cell lines: NCaP, PC3, and DU145. TMPRSS2 overexpression in these three cell lines had no effect on cell growth, but it was significantly associated with increased invasiveness. The latter is believed to be associated with the proteolytic activity of TMPRSS2, as DU145 cells transduced with a mutant version of TMPRSS2 with impacted protease activity did not demonstrate increased cellular motility [25]. Another study used the transgenic adenocarcinoma of the mouse prostate (TRAMP) model and found that TMPRSS2 expression was correlated with an increased frequency of gross metastasis. Additionally, the study suggested that TMPRSS2 plays a critical role in the enzymatic activation of the hepatic growth factor (HGF), which, in turn, promotes c-MET receptor tyrosine kinase signaling, leading to the enhancement of a pro-invasive epithelial-to-mesenchymal transition phenotype [19]. Moreover, in a study conducted by Wilson et al., TMPRSS2 was found to be able to catalytically activate PAR-2 in prostate cancer cell lines, leading to the increased expression of matrix metalloproteinases 2 and 9. These two molecules are known to degrade the extracellular matrix, enhancing cancer metastasis [18].

Another important aspect of the pathogenetic role of *TMPRSS2* in cancer is its involvement in chromosomal rearrangements. Chromosomal rearrangements were known to participate in the development of hematological malignancies and sarcomas, but their association with the pathogenesis of common solid carcinomas was largely unexplored. In a study performed by Tomlins et al., a recurrent chromosomal rearrangement between the *TMPRSS2* gene and two members of the E26 transformation-specific (ETS) family of transcription factors, namely the *ETS-related* gene (*ERG*) *and ETS variant transcription factor 1* (*ETV1*), was identified. These molecules were previously found to be overexpressed in prostate cancer, and their aberrant expression was linked to the occurrence of this genomic event [26]. Subsequent inquiries revealed that other members of the ETS family, such as *ETV4*, can also participate in molecular fusion with the *TMPRSS2*, albeit with a lesser frequency. Collectively, these findings imply that the overexpression of the ETS transcription factors, resulting from chromosomal rearrangements involving the *TMPRSS2* genetic elements, represents a crucial facet of the molecular pathology of prostate cancer [27].

The fusion of the promoter of *TMPRSS2* with the coding region of *ERG* is the most prevalent molecular aberration in prostate cancer; it occurs in approximately 50% of prostate cancer cases and is the most common gene fusion in solid tumors [28]. Both genes are located on chromosome 21, about 3Mb apart. The fusion is considered to occur early in the pathogenesis of prostate cancer and can be caused by either genomic translocation or interstitial deletion of the intergenic region between the two genes. The promoter region of the *TMPRSS2* contains androgen-sensitive elements, and as a result, this fusion sets the expression of *ERG* under hormonal control, promoting its overexpression in the settings of an androgen-rich environment [29,30,31].

Several studies tried to investigate the pathogenetic role of *TMPRSS2:ERG* fusion in carcinogenesis and the development of prostate cancer. Zhou et al. demonstrated that ERG promotes the expression of the α1 and β1 subunits of soluble guanylyl cyclase (sGC), which catalyzes the production of cyclic guanosine monophosphate (cGMP) in endothelial cells. The study highlights the fact that *TMPRSS2:ERG* fusion transcriptionally upregulates sGC; promoting cGMP synthesis in prostate cancer cells thus enhances cellular proliferation. The pro-proliferative activity of the sGC-cGMP pathway is potentially linked to the activation of PI3K/AKT signaling, which is a downstream target of the sGC-cGMP pathway, modulating cell survival, migration, and angiogenesis [32].

Deplus et al. developed cell lines of luciferase-expressing PC3M cells carrying the *TMPRSS2:ERG* fusion. In vitro analysis showed increased migration rates of cells carrying the fusion compared to control cells. In the same study, the researchers injected these two cell lines into murine models. They observed that the mice injected with the *TMPRSS2:ERG* positive cells showed a 57% higher incidence of bone metastasis. Interestingly, these metastases were predominantly located in the hind limbs and spine, which are the two most common anatomical sites affected by prostate cancer in humans. In addition, the authors conducted transcriptomic profiling to investigate the differences in gene expression between the fusion-positive and control cell lines. According to the results of this study, the overexpression of *TMPRSS2:ERG* was associated with the aberrant expression of the genes that enhance osteomimicry of cancer cells, facilitating their invasion and growth in bone tissue [33].

## 3. Clinical Implications of *TMPRSS2:ERG* Fusion in Prostate Cancer

Despite the fact that *TMPRSS2:ERG* fusion is a frequently occurring molecular event in prostate cancer, its precise impact on the clinical aspects of the disease remains controversial. Taris et al. investigated ERG expression throughout all stages of prostate cancer natural history, from high-grade prostatic intraepithelial neoplasia (HGPIN) to metastasis, in a multinational patient cohort. They observed that among Caucasian patients, ERG expression followed a positive trend from HGPIN (17.5%) to clinically localized prostate cancer (33%) and metastases (53%) (*p* = 0.01). Additionally, a higher level of ERG expression was correlated with a more advanced pathologic stage for tumors confined within the prostate gland (pT3 = 43%, pT2 = 27.5%). The authors also aimed to examine the prognostic importance of ERG expression in patients with prostate cancer who underwent surgical prostatectomy. They utilized a case–control population of 65 patients with recurrence and 65 patients without recurrence. Following adjustments for age, preoperative PSA, Gleason score, and pTNM stage, the study found a significant correlation between ERG expression and biochemical progression-free survival. This so-called “prognostic-paradox” regarding the correlation of *TMPRSS2:ERG* fusion with a more advanced disease and a more favorable prognosis may be attributed, at least in part, to the fact that the expression of *ERG* is dependent on the AR activation, indicating a potential for an improved response to hormone castration interventions [34].

Chalmers et al. sought to investigate the molecular aberrations that define early-onset prostate cancer, as a distinct and understudied clinical entity. Patients with early-onset prostate cancer exhibit a lower frequency of risk factors and are likely to possess multiple genetic mutations that dictate a more aggressive disease phenotype. The study analyzed comprehensive genomic profiling data from 10,189 prostate cancer patients of different racial and age groups. Interestingly, the incidence of *TMPRSS2* fusions was higher in patients aged ≤50 years than those aged ≥60 years, with a decreasing incidence with age, while other gene alterations tended to increase with age. The findings of this study suggest that a distinct group of young patients with aggressive prostate cancer is frequently associated with *TMPRSS2:ERG* fusions, while exhibiting a lower likelihood of harboring mutations in other genes, such as *AR*, *SPOP*, and *ASXL1* [35].

The detection of the *TMPRSS2:ERG* fusion gene in urine samples after digital rectal examination as a non-invasive procedure was studied over a decade ago. The results since then have been quite encouraging as the urine test featured, despite its low sensitivity of 37%, a specificity of 93% and a positive predictive value of 94% in post-DRE urine samples examined from men with PCa-positive and PCa-negative biopsies using semiquantitative reverse transcription PCR (RT -PCR). Indeed, when the detection of *TMPRSS2:ERG* fusion was combined with *PCA3* RNA transcripts, the sensitivity of the method was improved. Therefore, the combination could be used as a diagnostic marker in patients with indications of prostate cancer, such as elevated serum PSA values and a history of negative biopsy, directing the need for a repeat biopsy [36,37].

A few years ago, a meta-analysis demonstrated that ERG overexpression or positive fusion status was associated with the advanced pathological characteristics of patients with prostate cancer. More specifically, *TMPRSS2:ERG* fusion was more common in the T3–4 stages of PCa than in the T1–2 stages and in cases with distant metastasis (M1), whereas no difference was observed in the lymph node status. Moreover, the fusion gene was common in young patients aged ≤65, in patients with high PSA levels (>10 ng/mL), and in cases with peripheral involvement. The *TMPRSS2:ERG* fusion was not associated with biochemical recurrence [38].

Regarding the association with the Gleason score in the meta-analysis of Song et al., as well as that of Fine et al., the *TMPRSS2:ERG* fusion was more frequently associated with the lower scores (≤7) and was associated with less aggressive histological features of prostate cancer [38,39].

The association of ERG overexpression with specific histomorphologic features and prognosis in prostate cancer has been a topic of debate since the discovery of the *TMPRSS2:ERG* fusion. However, studies conducted worldwide have reported conflicting findings. Fine et al. conducted FISH analysis in a cohort of prostate cancer patients who underwent radical prostatectomy without neoadjuvant therapy study to investigate the correlation between *TMPRSS2:ERG* fusion and the Gleason score. The study revealed that the genomic rearrangements in *TMPRSS2:ERG* were associated with a lower Gleason score, while an increase in gene copy number was linked to a higher Gleason score [39].

In another study, conducted by Peterson et al., the expression of ERG protein, serving as an indicator of *TMPRSS2:ERG* gene fusion, was evaluated using immunohistochemical staining in a cohort of 1180 patients. In the same inquiry, the authors performed a meta-analysis of 47 studies to further investigate the association between gene rearrangement and the prognosis of the patients. The study revealed that 49% of the cohort exhibited ERG overexpression and that they were more likely to have a higher tumor stage; however, no significant correlation was observed between the ERG expression and the Gleason score or the clinical outcomes of the disease. These findings were supported by the concurrent meta-analysis, which revealed that the presence of *TMPRSS2:ERG* fusion increased the risk of a higher tumor stage at diagnosis but did not correlate with the final outcome [40].

Overall, *TMPRSS2:ERG* fusion is considered to be relevant for the disease. However, its correlation to prostate cancer development and progression, as well as its clinical significance, are not yet fully clarified. Due to its high frequency in prostate cancer cases, the *TMPRSS2:ERG* fusion might be more promising as a diagnostic marker rather than as prognostic [41].

Taxanes remain up to date in the main chemotherapeutic regimen in the treatment of metastatic castration-resistant prostate cancer (mCRPC). However, in preclinical studies the *TMPRSS2:ERG* rearrangement has been associated with taxane resistance. Reig et al. evaluated *TMPRSS2:ERG* expression in peripheral blood mononuclear cells and tumor tissue from mCPRC patients treated with taxanes. The results of this study indicate that the detection of *TMPRSS2:ERG* in blood from mCRPC patients treated with docetaxel is correlated with lower PSA response rate (12.5% vs. 68.3%, *p* = 0.005), as well as clinical/radiological-PFS (3.1 mo vs. 8.2 mo, *p* < 0.001) [42].

In an effort to evaluate the prognostic value of *TMPRSS2:ERG* in patients with mCRPC treated with enzalutamide, an early phase clinical trial described better PSA responses in *TMPRSS2:ERG-associated* tumors. However, the latest clinical trial phase II proved that the fusion has limited value as a predictive biomarker in mCRPC treated with anti-androgen therapies, such as abiraterone [43].

## 4. *TMPRSS2:ERG* Therapeutics

The *TMPRSS2:ERG fusion*, as one of the most common molecular aberrations in prostate cancer, has been extensively studied as a potential therapeutic target. In preclinical studies, several molecules that act as ERG antagonists have been developed through in silico methods. Currently, these drugs are not used in clinical practice and remain in preclinical studies with promising results [44].

## 5. Biological and Biochemical Features of KATs

### 5.1. Lysine Acetyltransferases (KATs) of AR

The CREB-binding protein (CBP) and p300 are two paralogous lysine acetyltransferases (KATs). Both molecules act as co-activators, interacting with other DNA-binding transcription factors to induce epigenetic modifications in the chromatin structure [45,46,47]. The CBP/p300 complex plays an important role in histone acetylation in promoters [48], enhancers [49,50], and super-enhancers [51,52,53] throughout the genome. Furthermore, some of the transcription factors that CBP/p300 acetylates are p53 [54], HIF-1α [55], c-Myb [56], and STAT-1 [57,58], with c-MYC [59] and GATA-1 [60] modifying their activity.

CBP and p300 exhibit functional redundancy [61], although they have distinct roles in specific situations. Both co-activators are necessary for proper embryonic development and differentiation in various cell types [62,63]. Due to their structural similarity and functional overlap, CBP and p300 are often mentioned together.

### 5.2. Location of Gene

The *EP300* gene is located on chromosome 22q13.2. The encoded protein P300 (also known as KAT3B) has a size of 300 kDa and is a lysine acetyltransferase (KATs) that was discovered in the 1980s–1990s [64,65,66]. This protein was originally identified by its interaction with E1A, an adenoviral-transforming protein [67] as it is the nuclear binding target of the E1A cancer protein.

On the other hand, the *CBP* gene is located on chromosome 16p13.3 and has been identified as the binding partner and co-activator of the cAMP response element-binding (CREB) protein [68].

### 5.3. Role of P300/CBP Complex

Due to its ability to acetylate histone and non-histone proteins [69,70,71], p300 regulates the transcription of genes. As a transcription factor, it regulates physiological processes such as DNA damage response, cell differentiation, proliferation, and apoptosis through its interactions with over 400 factors [72]. This protein also contributes to homeostasis and growth control, among other biological functions. Over the past two decades, research has proven that p300 plays an important role in nuclear hormone signaling pathways, which promote tumor growth in several cancer types [73].

As crucial co-regulators for many nuclear receptors, both molecules cause specific lysine residues in their substrate proteins to be acetylated. In addition to acetylating all four histones and various transcription factors, several nuclear receptors including thyroid hormone receptor (T3R), estrogen receptor (ER), retinoid X receptor (RXR), farnesoid X receptor (FXR), progesterone receptor (PR), AR, and steroidogenic factor 1 (SF-1), have shown to be direct substrates for p300 lysine acetyltransferase activity [74,75,76,77,78,79,80].

In light of their ability to alter epigenetic landscapes and protein functions, KAT proteins play a crucial role in the development of a variety of human diseases, including cancer, as a result of their deregulated expression and function. Therefore, drugs can be developed to target these proteins to treat a variety of diseases [81,82,83,84]. According to Jaiswal B et al. [85], various KATs modulate AR function and are thus regarded as potential therapeutic targets for prostate cancer.

### 5.4. p300 Protein Description

p300 contains conserved domains, including a central catalytic domain for acetylating proteins that is adjacent to a bromodomain and a PHD finger (CH2) for chromatin association and modification [86,87]. Among the four transactivation domains flanking the central domain are: (i) the cysteine–histidine rich region 1 (CH1), containing the transcriptional adapter zing finger 1 (TAZ1); (ii) a kinase-inducible interacting domain (KIX); (iii) another cysteine–histidine-rich region (CH3) containing a TAZ2 and a ZZ domain [88] that have been shown to interact with a variety of proteins; and (iv) a nuclear receptor co-activator binding domain (IBiD) [89].

As a result of its structure, p300 functions as a stabilizing scaffold between transcription machinery and transcription factors, which bind to its CH1, CH3, and KIX domains [90]. Through its histone acetylation domain (HAT) [91], p300 acetylates histones directly in order to promote transcription. Furthermore, it acetylates non-histone proteins and modifies their functions [92].

Classical models of p300/CBP as scaffolds were founded on experiments performed on the promoter of b-interferon in response to viral infection. As a result of stimulation, the b-interferon enhanceosome forms on the surface of the p300 molecule, bringing transcription factors and the RNA polymerase holoenzyme to activate the b-interferon gene in a rapid but effective manner [93].

### 5.5. Role of p300 in the Cell Cycle and Proliferation

p300 plays an important role in forming the transcription preinitiation complex, a large multiprotein complex whose function is to initiate gene transcription. This is accomplished partly by p300 auto-acetylation [70,94], which facilitates its dynamic association with and dissociation from the transcriptional machinery.

A key function of p300 is to maintain the cell type—specific gene expression, and therefore preserve cell identity [95]. More specifically, in Ebrahimi et al.’s research it is shown that by inhibiting p300 HAT activity, global histone deacetylation is induced, and pluripotent stem cells cannot be formed, indicating that HAT activity is required for many transcription factors. More than 16,000 genes in human cells are bound by p300, demonstrating its pervasive role in transcriptional regulation. Not all binding events result in transcriptional activation, and there is growing evidence that p300 plays a gene-repressive role in certain situations [96,97]. As well as influencing cell cycle progression, p300 acetylates or interacts with proteins involved in DNA and chromatin replication, which are essential for DNA replication [98].

### 5.6. P300 Induces Oncogene Transcription, Promotes Cancer Cell Proliferation, Survival, Tumorigenesis, Metastasis, and Immune Evasion

The recent research of Felice Ho-Ching Tsang and his team indicates that the overexpression of the *EP300* gene is positively correlated with gene copy number variations. Moreover, with poor prognosis in patients with hepatocellular carcinoma, where it significantly reprograms super-enhancers, it promotes overexpression of the oncogenes associated with super-enhancers, such as *MYC*, *MYCN*, and *CCND1*, and it facilitates tumor progression in vivo and cell proliferation in vitro [99].

In a different study, the knockdown of *EP300* in melanoma cells led to downregulation in 33 out of 250 genes that are targeted by the transcription factor MITF. In addition, p300 upregulated the expression of MITF target oncogenes, including *FOXM1*, by inducing H3K27ac at the promoter of the MITF gene. This resulted in an increased proliferation in melanoma cells [100].

A study published in 2019 characterized *EP300* as an oncogene that correlates with poor prognosis in esophageal squamous carcinoma [101]. In the same inquiry, the authors demonstrated that the mutations of *EP300* and its aberrant expression were associated with poorer prognosis and shorter survival. Finally, they showed that the knockdown of the gene reduced cellular proliferation, migration, and invasion in esophageal squamous carcinoma.

Xiao et al. correlated the high expression of p300 with more aggressive breast cancer. Moreover, they suggested that disease recurrence might be acquired through the high expression of p300 [102].

In colorectal cancer, the research of Wang et al. has shown that p300 acetylates PHF5A and in this way regulates KDM3A expression through alternative splicing to promote cell proliferation and tumorigenesis [103].

A loss of the tumor suppressor VHL induces p300 to recruit oncogene enhancers and super-enhancers in clear cell renal cell carcinoma. Consequently, p300 promotes the acetylation of H3K27ac, resulting in the upregulation of ccRCC-specific genes, including the master regulator ZNF395. The interaction between HIF2α, p300, and H3K27ac contributes to enhancer dysfunction and the development of ccRCC tumorigenesis. The above lead to overexpression of the oncogenes MYC, and ZNF395 and therefore renal clear cell carcinoma proliferation, survival, and colony formation in vitro and tumor progression in mice [104].

The research of Diesch J. et al. suggests that in the myelodysplastic syndrome (MDS)-derived acute myeloid leukemia cell, CBP/p300 promotes the expression of ribosomal genes, which are critical for protein synthesis. The inhibition of CBP/p300 reduces global protein synthesis, leading to leukemia cell death. This finding suggests that CBP/p300 may be a potential target for the development of new therapies for acute myeloid leukemia (AML) [105].

The recruitment of T regulatory cells and myeloid-derived suppressor cells to tumor tissues is an important mechanism of immune evasion by cancer cells. CBP/p300 plays a role in this process by inducing H3K27 acetylation at the promoters and enhancers of the genes critical for T regulatory cells and myeloid-derived suppressor cells. CBP/p300 upregulates the expression of these genes, thereby promoting T regulatory cell and myeloid-derived suppressor cell survival and function and suppressing cytotoxic T cell-driven immunity, lymphocyte activation, and proliferation. This finding suggests that CBP/p300 inhibitors may be useful in the development of new cancer immunotherapies that target T regulatory cells and myeloid-derived suppressor cells [106,107,108,109].

### 5.7. The Transcriptional Acts of p300 in Prostate Cancer

In order to assemble a transcription complex on the promoter of an AR target gene in response to dihydrotestosterone, it is necessary to recruit specific co-regulators (such as p300), modify histones, and recruit basal transcription factors and RNA polymerase II.

ChIP analysis performed by Ianculescu I. et al. demonstrated that p300 is recruited to the promoter in response to DHT. The analysis of the mRNA levels of the AR target genes following p300 depletion confirms that p300 regulates a wide range of AR target genes [110].

More specifically, on the *TMPRSS2* gene, even though the presence of p300 is not mandatory for the DHT-induced binding of AR to ARE, every other DHT-induced event is dependent on p300. To be more specific, there is a sequence of these events that includes the increase in the expression of H3, H4 histones and H3K4 methylation, as well as the recruitment of the TATA-binding protein and RNA polymerase II. In addition to its role, p300 is needed for the recruitment of the basal transcription machinery, particularly TBP and RNA polymerase II.

TBP is considered a critical component of the transcription factor IID (TFIID) complex that is required for all transcriptional work by RNA polymerase II. Due to p300’s interaction with TBP [111], it is possible that p300 recruits TBP directly. It is also possible that p300 indirectly recruits TBP by acetylating histones or recruiting other co-regulators.

In order to understand how p300 regulates androgen-regulated genes, we consulted the article of Sawant M. et al. In this article, we discovered that p300 recognizes phosphorylated AR, which promotes the subsequent AR acetylation at K609, thereby facilitating transcription [112].

As reported by Ianculescu I. and her team, in response to DHT-activated AR binding to the ARE a group of co-regulators, including p300, is recruited to the ARE as well [110]. Also recruited are co-activators such as p160, which serve as scaffolds for the recruitment of many other co-regulators. In the region of the ARE and the transcription start site, the recruitment of p300 increases H3K18ac, H3K27ac, and H4ac. Furthermore, there is an increase in H3K4me1 near the ARE and another one in H3K4me3 near the TSS. Finally, when TBP and Pol II are recruited to the TSS, a transcriptional response occurs.

### 5.8. p300 and Prostate Cancer

Many researchers have correlated the implication and impact of p300 in prostate cancer [7,85,113]. Under physiological conditions, in order to form a productive transcriptional AR complex, co-activators facilitate DNA occupancy, chromatin remodeling, and recruitment of the general transcription factors associated with the RNA polymerase II holocomplex and ensure that the AR proteins are stable, folded correctly, and/or distributed appropriately within the cell.

During PCa progression, a subset of these co-activators is overexpressed, and this overexpression significantly contributes to the activation of the AR through androgen depletion-independent (ADI) mechanisms.

Several analyses have demonstrated that high levels of p300 in biopsies predict larger tumor volumes, extraprostatic extension of disease, and seminal vesicle involvement in prostatectomy and can possibly be associated with prostate cancer progression after surgery [114]. Furthermore, an elevated expression level of p300 detected on patients’ tissue samples, via the use of immunohistochemistry, was positively correlated with a higher Gleason score and an aggressive prostate tumor type. More specifically, the p300 expression in high-grade tumors (Gleason score ≥ 7) was significantly higher compared to that of low-grade tumors (Gleason score < 7) [17.7% versus 13.7%, respectively, *p* = 0.03]. In the same study, patients with a high Gleason score and p300 expression had an increased risk of elevated PSA, indicating a biochemical recurrence (*p* = 0.002). Pathologists have also described a higher expression of p300 in undifferentiated prostate cancer [5].

CBP and p300 are co-activators of the AR and enhance its activity and signaling. The high expression of CBP and p300 is associated primarily with the AR signature in prostate cancer. The use of mRNA sequencing from biopsies of prostate cancer revealed that the expression of these molecules is associated with AR mRNA expression in early prostate cancer as well as in metastatic castration-resistant prostate cancer. Moreover, the high expression of p300 is significantly associated with a previously described acquired androgen deprivation therapy (ADT) resistance signature in primary PC and mCRPC. The use of immunochemistry in biopsies of castration-sensitive prostate cancer proved that neither the nuclear CBP protein nor the nuclear p300 protein expression was significantly (*p* = 0.28) altered as the prostate tumors progressed from castration-sensitive to castration-resistant prostate cancer. However, a positive association of the expression of nuclear CBP and p300 protein in castration-resistant prostate cancer has been noted [85,113].

Notably, p300 is upregulated in tumor samples from PCa patients treated with docetaxel. Docetaxel is a chemotherapeutic regimen, acting as a microtubule inhibitor and is widely used for treating CRPC. An elevated p300 mRNA expression in the tissues of the metastatic castration prostate cancer of patients treated with docetaxel was noticed. Patients who are under treatment with docetaxel experience resistance to the chemotherapy. In preclinical models, using docetaxel-resistant cell lines, an increase in p300 protein levels compared to those of the docetaxel-sensitive group has been noticed. The docetaxel-resistant PCa cell lines of p300 knockdown impair the clonogenic growth of these cell lines, suggesting that p300 may play a role in resistance to docetaxel. Targeting p300 in docetaxel-resistant PCa cell lines reduces the metastatic potential of docetaxel-resistant cells by reducing colony formation, migration, and invasion capability [115].

Figure 1 illustrates the molecular cross-play between p300 and *TMPRSS2:ERG* expression in hormone-dependent prostate cancer cells. The current evidence indicates that AR (AR) activation plays a critical role in the progression of the disease, even when androgen levels are low. This is attributed to the alternative activation mechanisms involving other molecular pathways, such as tyrosine kinase receptor activation [116,117,118].

## 6. The Use of p300 as a Therapeutic Target in Prostate Cancer

Prostate carcinogenesis is driven by activation of the AR, which functions as a transcription factor which regulates the expression of multiple genes that are key to the growth and proliferation of the cancer cells. Therefore, the therapies that target AR signaling have been successful in improving the outcome of patients with castration-sensitive and castration-resistant prostate cancer (CRPC). However, within a few years, therapy fails, and the patient develops drug resistance leading to the spread of androgen-independent PCa or castration-resistant PCa. The development and progression of CRPC is characterized by continued AR signaling, either through overexpression of the *AR* by gene amplification; mutations of the AR ligand binding domain (LBD); tumor-derived androgen production; expression of constitutively active AR splice variants (AR-SVs), of which AR-V7 remains the best studied; or co-regulator overexpression, including the transcription factor co-activator proteins p300/CBP [45,119]. Treatment options for non-metastatic and metastatic CRPC include inhibitors of androgen synthesis and anti-androgens such as enzalutamide and apalutamide, as well as the chemotherapeutic compound docetaxel [80].

Several preclinical studies have shown the benefit of the use of inhibitors of p300/CBP in in vitro and in vivo models, but to date, none of these molecules is used in clinic. Two p300/CBP inhibitors are currently in clinical trials in cancer patients. CCS1477 was developed as a potent, selective, and orally bioavailable inhibitor of the conserved bromodomain of p300 and CBP and is currently the first p300/CBP inhibitor in phase I/IIa clinical trials in patients with advanced solid tumors. (ClinicalTrials.gov Identifier: NCT03568656). In cell line models of castration-resistant prostate cancer, CCS1477 inhibited AR signaling by reducing the expression of AR-regulated genes (*KLK2*, *KLK*, and *TMPRSS2*), whereas in a mouse xenograft model it managed to inhibit tumor growth and AR signaling by reducing AR-FL and AR-V7 protein expression in all the dosing groups, as it did with C-MYC. The trial is evaluating the safety and efficacy of the drug as a monotherapy and in combination (with abiraterone acetate and enzalutamide). The primary outcome involves the incidence of treatment-related adverse events, whereas the secondary outcomes involve the biochemical PSA response and the circulating enumeration of tumor cells, the objective response rate (ORR), the radiological progression-free survival (rPFS) and the definition of AUC, and the maximum observed plasma concentration (Cmax) of CCS1477 [120].

The second drug, FT-7051, is an oral, potent, and selective inhibitor of p300/CBP with activity in preclinical models of prostate cancer, including models which are resistant to the currently used AR inhibitors such as enzalutamide. FT-7051 has entered a multi-center, phase I, open-label clinical trial in patients with metastatic castration-resistant prostate cancer who have progressed despite prior therapy and who have been treated with at least one potent anti-androgen (ClinicalTrials.gov Identifier NCT04575766). The trial will evaluate the safety and tolerability of FT-7051 and determine the recommended phase 2 dose (RP2D) as well as the pharmacokinetics (PK), preliminary anti-tumor activity, and pharmacodynamics (PD) of the drug. The results from the trials are expected to be announced soon [121].

Table 1 lists some of the p300/CBP inhibitors that are currently being tested in preclinical and clinical trials in preclinical models and cancer patients, respectively [120,121].

## 7. The Interplay between P300 and TMPRSS2 in Prostate Cancer

To gain deeper insights into the role of p300 in regulating the *TMPRSS2* gene in prostate cancer, we analyzed the study conducted by Ianculescu et al. [110].

The research highlights the critical involvement of p300 in *TMPRSS2* gene expression in response to dihydrotestosterone (DHT), a hormone implicated in prostate cancer progression. This finding emphasizes the significance of p300 in the regulation of *TMPRSS2*, particularly in the context of DHT signaling. Furthermore, the study reveals that the depletion of p300 results in a significant reduction in DHT-induced *TMPRSS2* expression. This implies that p300 is necessary for the full activation of *TMPRSS2* gene expression in the presence of DHT. The depletion of p300 likely disrupts the molecular processes or signaling pathways required for the efficient transcription of *TMPRSS2* in response to DHT.

Additionally, the research highlights the significant role of p300 in DHT-induced chromatin remodeling at the *TMPRSS2* gene. It demonstrates that p300 plays a crucial role in facilitating the increase in histone acetylation of H3 and H4, which are associated with gene activation, suggesting that p300 is involved in promoting the activation of the *TMPRSS2* gene by facilitating histone acetylation. Moreover, the study demonstrates that p300 is essential for the methylation of histone H3 at lysine 4 (H3K4), another modification associated with active gene transcription, which further supports the involvement of p300 in regulating the transcriptional activity of the *TMPRSS2* gene in response to DHT.

The study also elucidates the role of p300 in the assembly of the transcription complex at the *TMPRSS2* gene. It reveals that p300 participates in the recruitment of transcription factors and the essential components of the transcription machinery, such as TBP and RNA polymerase II, which are required for transcription initiation.

Additionally, the study highlights the differential requirement of p300 for gene expression, specifically focusing on the *TMPRSS2* and *FKBP5* genes, both regulated by androgens. While p300 is recruited to both genes in response to DHT, its depletion selectively affects the DHT-induced expression of *TMPRSS2* but not *FKBP5*. This discrepancy is attributed to variances in the promoter environment and the presence of specific transcription factors or co-regulators at each gene.

These findings provide valuable insights into the functional role of p300 in the transcriptional regulation of the *TMPRSS2* gene. By facilitating the recruitment of the transcription factors and key components of the transcription machinery, p300 contributes to the efficient initiation of transcription at the *TMPRSS2* gene locus. Understanding the specific mechanisms and differential requirements of p300 in gene regulation can enhance our understanding of prostate cancer progression and potentially lead to the development of targeted therapies [110].

In a similar vein, we thoroughly examined the research conducted by Chen et al. [122]. The study investigates the regulatory mechanisms involving the prostate cancer-associated long noncoding RNA *PRCAT38* and the oncogene *TMPRSS2*.

The study unveils a direct regulation of the *TMPRSS2* gene by enhancer E1, which is responsive to DHT treatment, in the context of chromatin looping and enhancer–promoter interactions. Additionally, enhancer E2 serves as a “bridge” between enhancer E1 and the promoter of the *PRCAT38* gene, indirectly regulating *PRCAT38* through E1–E2 looping. The interaction frequencies of these enhancers with the promoters are regulated by the AR, FOXA1 binding, and the activity of p300. This indicates that p300 plays a role in the chromatin looping and enhancer-mediated gene regulation of *TMPRSS2* and *PRCAT38*.

Additionally, the involvement of enhancers E1 and E2 in the regulation of *TMPRSS2* and *PRCAT38* indicates a co-regulatory mechanism between these two genes, which is supported by the observation that their knockout leads to the downregulation of both genes. This finding provides insight into the co-expression of *TMPRSS2* and *PRCAT38* in prostate tissues. Moreover, the participation of p300 underscores its role in coordinating the expression of *TMPRSS2* and *PRCAT38*, further emphasizing its significance in this co-regulatory mechanism.

The study emphasizes the fact that the AR predominantly binds to enhancer regions rather than promoters. Upon AR binding, it facilitates the recruitment of FOXA1, p300, and Pol II, resulting in the activation of the enhancers. Consequently, there is an increased enrichment of Pol II and H3K27ac, an active enhancer-associated histone modification, at the promoter regions of *TMPRSS2* and *PRCAT38*. Although AR does not directly bind to the *PRCAT38* promoter, the looping interaction between enhancers and promoters potentially facilitates the recruitment of transcriptional machinery to the promoter region.

Overall, p300 plays a crucial role in mediating chromatin looping, enhancer activation, and co-regulation of *TMPRSS2* and *PRCAT38* in response to androgen signaling in prostate cancer cells. Its involvement in these processes suggests that targeting p300 could potentially impact the expression of both genes and could have therapeutic implications in prostate cancer treatment [122].

Lastly, to gain a better understanding of the correlation between the p300 and *TMPRSS2* genes, we refer to a relevant article from a clinical perspective.

The study of Welti et al. [113] suggests that p300 plays a role in the regulation of the *TMPRSS2* gene by being associated with AR signaling, acquired ADT resistance, and the growth of CRPC tumors. The study provides evidence that p300 is necessary for the expression of *AR* and *C-MYC*, and its inhibition through a bromodomain inhibitor can affect the expression of genes involved in AR signaling, potentially including *TMPRSS2*.

## 8. Conclusions

Despite the high incidence of prostate cancer affecting millions of individuals worldwide, the complete understanding of the molecular pathogenesis of this disease remains elusive. The identification of the *TMPRSS2:ERG* fusion as a frequent molecular alteration occurring in nearly half of prostate cancer cases has revolutionized our comprehension of the underlying mechanisms driving this malignancy. Multiple studies have indicated that TMPRSS2 has the potential to be a valuable biomarker, mainly for the diagnosis of prostate cancer. Further studies are needed in order to clarify the clinical significance of this molecule.

p300 represents another promising molecule that can be used as a prognostic marker, mainly in CRPC, as well as a predictive marker for the response to treatment with docetaxel. However, further studies are needed in order to provide solid recommendations. p300/CBP inhibitors are novel anticancer agents that have proven evidence of efficacy in the treatment of prostate cancer in preclinical models. As long as advanced PC remains fatal, clinical trials evaluating the efficacy of these drugs are needed, and improved therapeutic strategies remain an urgent unmet medical need.

In our opinion, this review presents several noteworthy positive aspects. Firstly, it comprehensively addresses various facets of TMPRSS2 and p300 biology, including their basic biochemical features and physiological functions and their implication in carcinogenesis in general as well as their important role in the molecular pathology of prostate cancer. Recent investigations highlighting the potential clinical utility of these two molecules as either diagnostic and prognostic biomarkers for prostate cancer or as therapeutic targets have also been incorporated. In our assessment, we have observed a dearth of studies investigating the dynamic interplay between TMPRSS2 and p300 in prostate cancer. This deficiency represents the primary limitation of our study. Given the scarcity of high-quality research pertaining to this issue, our ability to draw firm conclusions and uncover substantial findings regarding the potential clinical application of TMPRSS2 and p300 in the management of prostate cancer patients is influenced by bias.

## Figures and Tables

**Figure 1 ijms-24-11299-f001:**
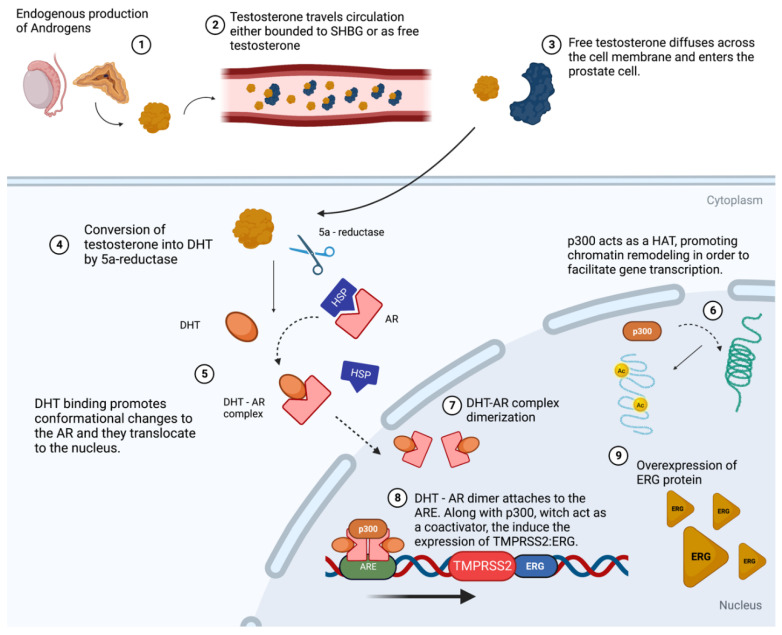
AR pathway in hormone-dependent prostate cancer cells. Under normal physiological conditions, the synthesis of androgens takes place in the Leydig cells of the testicles and in the zona reticularis of the adrenal gland. Testosterone, the prototypical androgenic steroid hormone, circulates in the bloodstream, bound to sex hormone-binding globulin (SHBG) or as free testosterone. Upon reaching the prostate cell, testosterone dissociates from SHBG and transverses the cellular membrane via diffusion. The enzymatic conversion of testosterone to DHT is mediated by 5a-reductase. Following binding with DHT, the androgen receptor (AR) dissociates from the heat shock protein (HSP). The DHT-AR complex undergoes conformational changes and is translocated within the nucleus, where it dimerizes. p300 acts as histone acetyltransferase (HAT) promoting the formation of euchromatin, which facilitates the binding of the DHT-AR dimer to the androgen-responsive elements (ARE) in the promoter region of the *TMPRSS2* gene. In addition, p300 functions as a co-activator, enhancing gene transcription and resulting in the overexpression of the *ERG* oncogene. Created with BioRender.com. (Created on 30 March 2023).

**Table 1 ijms-24-11299-t001:** Clinical trials for p300.

Clinical trial	Phase (Participants)	Disease	Drug	Primary Outcome Measures	Secondary Outcome Measures
NCT04575766	I (25)	Metastatic Castration-Resistant Prostate Cancer	FT-7051	DLTs AEs Clinical laboratory abnormalities	PSA response rTTP ORR CRR AUC Cmax Tmax T1/2
NCT03568656	I/IIa (350)	Metastatic Castration-Resistant Prostate Cancer	CCS1477 ± abiraterone acetate OR ± enzalutamide	Treatment-related AEs Laboratory assessments	PSA response ORR rPFS Cmax AUC
NCT05488548	I (50)	Castration-Resistant Prostate Cancer	EP31670	MTD DLTs RP2D	-

DLTs: dose limiting toxicities, AEs: adverse events, PSA: prostate-specific antigen, rTTP: time to radiographic progression, ORR: overall response rate, CRR: complete response rate, AUC: area under the plasma concentration, Cmax: peak plasma concentration, Tmax: time of peak plasma concentration, T ½: terminal elimination half-life, rPFS: radiological progression-free survival, MTD: maximum tolerated dose, RP2D: recommended phase 2 dose.

## Data Availability

Not applicable.

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
