# Peer review of "Unravelling the Role of P300 and TMPRSS2 in Prostate Cancer: A Literature Review"

_ijms, 2023, doi:10.3390/ijms241411299_

Round 1
Reviewer 1 Report (Previous Reviewer 1)
Thanks for your careful revision.
Author Response
Cover letter regarding the updated version of manuscript entitled as “Unravelling the role of P300 and TMPRSS2 in Prostate Cancer: a literature review”
Dear editor,
I hope this letter finds you well.
The purpose of this cover letter is to provide a comprehensive explanation of the revisions made to the manuscript, addressing each point in response to the insightful feedback received from the two esteemed reviewers.
I am writing to submit the revised version of our manuscript titled "Unravelling the role of P300 and TMPRSS2 in Prostate Cancer: a literature review." We appreciate the time and effort invested by the two reviewers in evaluating our work. Based on their valuable feedback, we have made several revisions that have significantly improved the quality and clarity of our article.
Here are the specific revisions we made in response to comments of the reviewers:
- Search Strategy, Inclusion, and Exclusion Criteria: We have incorporated the suggested additions to our manuscript by including a comprehensive description of our search strategy, inclusion criteria, and exclusion criteria. This addition enhances the transparency of our methodology and enables readers to understand the selective approach we employed in choosing relevant studies for our review.
- Revision of title in section 2.3: We have carefully considered the reviewer's comment regarding the redundancy of the term "cancer" in the title of section 2.3, which discusses the role of TMPRSS2 in prostate cancer carcinogenesis. Accordingly, we have removed the word "cancer" from the title, ensuring it accurately reflects the content of the section while maintaining conciseness.
- Rewording of Conclusions: In response to the reviewer's suggestion, we have reworded the last paragraph of our manuscript by removing the word "elucidate." We acknowledge the reviewer's point that this word is better suited for research articles and have made the necessary adjustments to ensure clarity and appropriateness in our conclusions. Furthermore, an additional paragraph has been included in the conclusions.
We believe that these revisions have significantly strengthened our manuscript. By incorporating the reviewers' suggestions, we have addressed important aspects such as methodological transparency, title clarity, and appropriate wording in the conclusions.
We would like to express our gratitude to the reviewers for their valuable input, which has undoubtedly improved the quality of our work. We remain committed to ensuring that our manuscript meets the high standards set by IJMS.
Thank you for considering our revised manuscript. We are confident that our research will contribute meaningfully to the field of prostate cancer and its molecular mechanisms. We appreciate the opportunity to publish our work in IJMS.
Should you require any further information or have additional suggestions for improvement, please do not hesitate to contact me. We look forward to the possibility of seeing our manuscript published in IJMS.
Thank you for your time and consideration.
Sincerely,
Charitomeni Gioukaki
Reviewer 2 Report (New Reviewer)
This manuscript covers many important aspects relating to this topic, however, my strong suggestion is to make it clear what search engine(s) and what search terms were used to determine which articles to include mention of and cite. Any reasons for article exclusion should also be stated. Doing this will help readers see that the review is comprehensive and unbiased. Other minor concerns include:
· Section 2.3: the word ‘cancer’ is redundant in the title ‘… prostate cancer carcinogenesis’. Should simply say prostate carcinogenesis.
· The first line of the last paragraph (conclusions) needs to be reworded. I don’t think it’s appropriate to say that you ‘aimed to elucidate…’ – this is a review paper, not a research paper. Elucidate is the right word to use – this paper is summarizing what is known.
Some awkward phrasing and inappropriate usage of wording but mostly ok
Author Response
Cover letter regarding the updated version of manuscript entitled as “Unravelling the role of P300 and TMPRSS2 in Prostate Cancer: a literature review”
Dear editor,
I hope this letter finds you well.
The purpose of this cover letter is to provide a comprehensive explanation of the revisions made to the manuscript, addressing each point in response to the insightful feedback received from the two esteemed reviewers.
I am writing to submit the revised version of our manuscript titled "Unravelling the role of P300 and TMPRSS2 in Prostate Cancer: a literature review." We appreciate the time and effort invested by the two reviewers in evaluating our work. Based on their valuable feedback, we have made several revisions that have significantly improved the quality and clarity of our article.
Here are the specific revisions we made in response to comments of the reviewers:
- Search Strategy, Inclusion, and Exclusion Criteria: We have incorporated the suggested additions to our manuscript by including a comprehensive description of our search strategy, inclusion criteria, and exclusion criteria. This addition enhances the transparency of our methodology and enables readers to understand the selective approach we employed in choosing relevant studies for our review.
- Revision of title in section 2.3: We have carefully considered the reviewer's comment regarding the redundancy of the term "cancer" in the title of section 2.3, which discusses the role of TMPRSS2 in prostate cancer carcinogenesis. Accordingly, we have removed the word "cancer" from the title, ensuring it accurately reflects the content of the section while maintaining conciseness.
- Rewording of Conclusions: In response to the reviewer's suggestion, we have reworded the last paragraph of our manuscript by removing the word "elucidate." We acknowledge the reviewer's point that this word is better suited for research articles and have made the necessary adjustments to ensure clarity and appropriateness in our conclusions. Furthermore, an additional paragraph has been included in the conclusions.
We believe that these revisions have significantly strengthened our manuscript. By incorporating the reviewers' suggestions, we have addressed important aspects such as methodological transparency, title clarity, and appropriate wording in the conclusions.
We would like to express our gratitude to the reviewers for their valuable input, which has undoubtedly improved the quality of our work. We remain committed to ensuring that our manuscript meets the high standards set by IJMS.
Thank you for considering our revised manuscript. We are confident that our research will contribute meaningfully to the field of prostate cancer and its molecular mechanisms. We appreciate the opportunity to publish our work in IJMS.
Should you require any further information or have additional suggestions for improvement, please do not hesitate to contact me. We look forward to the possibility of seeing our manuscript published in IJMS.
Thank you for your time and consideration.
Sincerely,
Charitomeni Gioukaki
This manuscript is a resubmission of an earlier submission. The following is a list of the peer review reports and author responses from that submission.
Round 1
Reviewer 1 Report
Thanks for having me review the manuscript “The dynamic interplay between P300 and TMPRSS2 in prostate cancer: a literature review” submitted to the journal “IJMS”. In this study, Charitomeni et al. went through the published scientific articles and tried to elucidate the molecular impact of p300 and tmprss2 in prostate cancer. This well-designed review would benefit researchers who are not familiar with this topic. However, some small questions should be addressed.
1. Page 2 line 48, add the full name of PSA and check if all the full names of abbreviations are provided at their first appearance.
2. Page 2 line 51, please add references after “…in regulating cancer progression.”
3. Page 2 line 62, the full name of AR appears above.
4. Page 3 line 97, did you analyze the database online eg. TCGA, to see if TMPRSS2 is dysregulated compared to the normal tissue? Please add these results here.
5. Page 3 line 107, in vitro should be italic; please check it throughout the manuscript.
6. Page 7 line 307, in part 12, “immune evasion” has not been discussed.
7. Page 8 line 389, please delete the blank between p and 300.
8. Page 9 line 409, please add references after “… docetaxel-sensitive group has been noticed.”
9. part 16 “Conclusion” should also contain TMPRSS2 contents.
Author Response
Dear Reviewer,
This cover letter aims to explain, point by point, the details of the revisions to the manuscript, in response to the comments provided by the two acclaimed reviewers.
- Issue regarding abbreviations
We have thoroughly gone through our text and tried to correct abbreviations, reassuring that the full names are only provided at their first appearance.
- Issue regarding reference and citations
We have added references where they were missing. Regarding the first reviewer’s comment about the citation from page 2 line 51, after “…in regulating cancer progression.” (Indicated as references number 7,8). Regarding the first reviewer’s comment about the citation from page 9 line 410 after “… docetaxel-sensitive group has been noticed.” the corresponding reference is marked at the end of the paragraph (indicated as reference number 115). Regarding the second reviewer’s comment about the citation from Kim et al. (line 86) the corresponding reference was marked at the end of the paragraph (indicated as reference number 20).
- Rephrasing the scientific methods in the abstract.
In response to the comment made by the second reviewer pertaining to the integration of up-to-date articles obtained from online journals and websites, we have modified our phrasing. Specifically, we have limited our citations to solely encompass peer-reviewed articles. Our reference to "websites" was intended to convey the utilization of online libraries and search engines, such as Pubmed and Google Scholar, as investigative tools to discern the most pertinent publications.
- “ETS or EGR: the fusion of TMPRSS2 is with ERG, but the abstract includes ERG and ETS.”
Regarding the above comment made by the second reviewer, the TMPRSS2 fusion occurs with different members of the ETS family, although its most prevalent association is with ERG. Since we have mostly included studies that have investigated the pathogenic role of TMPRSS2:ERG fusion in prostate cancer, we decided to refer only to the fusion with ERG in the abstract.
- Use of Figure 1 and Table 1 as supplementary materials
The two graphs were meant to be included as in-text materials and they were mistakenly included as supplementary during the upload of the manuscript.
- Reformatting of the Conclusion section
We have reformatted the conclusion section to incorporate a summary of the findings regarding the involvement of TMPRSS2 in prostate cancer. Additionally, we have included a final paragraph to highlight the interaction between TMPRSS2 and P300 within prostate cancer cells and their role in promoting cancer development and progression.
- “Immune evasion” has not been discussed (Page 11 line 307, in part 12)
We have added some extra context about immune evasion.
- Quality of English language
We have kindly asked a native English speaker to associate to read the text and give comments about the use of language and we have included them in our manuscript.
- “Page 3 line 97, did you analyze the database online eg. TCGA, to see if TMPRSS2 is dysregulated compared to the normal tissue? Please add these results here.”
Regarding the above comment made by the first reviewer, our comment about the dysregulated expression of TMPRSS2 in malignant tissue was extracted from the scientific articles cited below in the text.
- About the title
As we agree with the second’s reviewer opinion, we decided to reformat our title to improve its correlation with the context of our text in order to make it more representative.
So, our new title will be:
Unravelling the role of P300 and TMPRSS2 in Prostate Cancer: a literature review
We hope that our explanations are adequate and descriptive enough. Please do not hesitate to contact us for any further clarifications or corrections. We remain at your disposal for any further.
Finally, we would like to express our gratitude towards the two reviewers for their valuable feedback, which has greatly improved our manuscript. Their constructive criticism has strengthened our work, and we appreciate their time and effort in reviewing our work.
Reviewer 2 Report
This manuscript reviews the role of p300 and TMPRSS2 in prostate cancer. The title states that the main focus is the dynamic interplay between p300 and TMPRSS2. However, a description is hardly given of their dynamic interplay.
Major
· The interplay between p300 and TMPRSS2: The main message should be the interplay between p300 and TMPRSS2 based on the title. The last sentence in the Abstract states, “Overall, the interplay between the p300 and TMPRSS2-ETS fusion pathways is an active area of research.” However, the main body does not include a comprehensive review of their interplay.
· ETS or EGR: the fusion of TMPRSS2 is with ERG, but the abstract includes ERG and ETS.
· TMPRSS2-ERG fusion: the fusion was stated as frequent, but the effect of the fusion was not adequately described.
Minor
· Figure 1 and Table 1: They are listed as supplementary materials, but they are in the main text and not treated as supplementary materials.
· Website: The abstract states the references from online journals and websites. It is not appropriate to cite the materials on the website rather than citing the peer-reviewed published articles.
· Line 86: the reference citation is needed from Kim et al.
· AR: androgen receptor was fully written at many places, but AR was also used at other places. AR should be defined first, and then AR should be used instead of “androgen receptor.”
The quality of English is fine, but proofreading is necessary.
Author Response

(The authors gave the same response as above.)

Round 2
Reviewer 2 Report
The responses to the three major comments are not satisfactory. The authors apparently did not carefully read or understand the comments. The overall quality is poor and not appropriate for publication in IJMS.
Overall, it is readable.